# Adapting population health interventions for new contexts: qualitative interviews understanding the experiences, practices and challenges of researchers, funders and journal editors

Lauren Copeland [iD],[1] Hannah J Littlecott,[1,2] Danielle Couturiaux,[1] Pat Hoddinott [iD],[3] Jeremy Segrott [iD],[1,4] Simon Murphy,[1] Graham Moore,[1] Rhiannon E Evans [iD] [1]

**Correspondence to**
Dr Lauren Copeland;
copelandlc@cardiff.ac.uk

## ABSTRACT

**Objectives** Research on the adaptation of population health interventions for implementation in new contexts is rapidly expanding. This has been accompanied by a recent increase in the number of frameworks and guidance to support adaptation processes. Nevertheless, there remains limited exploration of the real-world experiences of undertaking intervention adaptation, notably the challenges encountered by different groups of stakeholders, and how these are managed. Understanding experiences is imperative in ensuring that guidance to support adaptation has practical utility. This qualitative study examines researcher and stakeholder experiences of funding, conducting and reporting adaptation research.

**Setting** Adaptation studies.

**Participants** Participants/cases were purposefully sampled to represent a range of adapted interventions, types of evaluations, expertise and countries. Semistructured interviews were conducted with a sample of researchers (n=23), representatives from research funding panels (n=6), journal editors (n=5) and practitioners (n=3).

**Measures** A case study research design was used. Data were analysed using the framework approach. Overarching themes were discussed within the study team, with further iterative refinement of subthemes.

**Results** The results generated four central themes. The first three relate to the experience of intervention adaptation (1) involving stakeholders throughout the adaptation process and how to integrate the evidence base with experience; (2) selecting the intervention and negotiating the mismatch between the original and the new context; and (3) the complexity and uncertainty when deciding the re-evaluation process. The final theme (4) reflects on participants' experiences of using adaptation frameworks in practice, considering recommendations for future guidance development and refinement.

**Conclusion** This study highlights the range of complexities and challenges experienced in funding, conducting and reporting research on intervention adaptation. Moving forward, guidance can be helpful in systematising processes, provided that it remains

---

### STRENGTHS AND LIMITATIONS OF THIS STUDY

⇒ The methodology captured a diverse and nuanced range of perspectives in relation to intervention adaptation.

⇒ The sampling ensured that we captured a wide range of studies including micro, meso and macro level interventions which allowed us to explore adaptation research experiences.

⇒ The primary limitation of this study was that we were unable to recruit patient and public involvement (PPI) and policy makers, limiting diversity in the perspectives reflected in our data.

⇒ Without the input from policy makers and PPI, the study lacks insight into how intervention adaptation is commissioned and resourced at a national and local level and how adaptation is understood by PPI contributors.

---

responsive to local contexts and encourage innovative practice.

---

## BACKGROUND

Research on the adaptation of population health interventions for implementation in new contexts is evolving at speed.[1–7] Adaptation is when intentional changes are made to an evidence informed intervention, either proactively or in response to emerging challenges, in order to improve the contextual fit within a new setting. This evolution accompanies the increased recognition that intervention effects do not always directly transfer to new contexts[4 5 8–10] and that adapting an existing intervention may be more efficient than de novo intervention development.[11] Within the ADAPT study population health interventions are defined as interventions or policies in public health or health services

that aim to change the population distribution of risk at either the micro, meso or macro level.[12]

In response to the emerging research on adaptation, there has been a significant increase in frameworks and guidance to support these processes.[1 2 13–15] While a number of these frameworks are explicitly grounded in empirical examples of adaptations, they often provide limited exploration of the real-world practice of undertaking adaptation, notably the complexity and challenges encountered by a diverse range of stakeholder groups.[16] Equally, stakeholder involvement and coproduction has been increasingly recognised as imperative in the complex process of development, adaptation and evaluation of interventions.[14 17 18] This, however, is something that has been underexamined in relation to adaptation. Furthermore, there has been limited research exploring the uptake and usefulness of guidance and frameworks to support adaptation which is important given that it seems to be rarely used. Publication of existing guidance has been relatively recent, which may explain the limited reports of guidance use and impact. It is important to consider how frameworks have been, and might be, integrated into real world practice to maximise their impact.[19]

This qualitative study examines stakeholders' experiences of funding, conducting and reporting of adaptation of complex interventions. It aims to understand the complexities and the practical challenges of conducting adaptation research. It was undertaken concurrently with other work packages as part of the ADAPT study (2018–2020), which aimed to develop evidence and consensus-informed guidance[12] that was grounded in the theoretical, methodological and real-world understandings, experiences and perspectives of a diverse range of relevant stakeholders.

### The ADAPT study

The ADAPT study (2018–2020) was funded by the UK MRC-NIHR methods panel to develop population health interventions' adaptation guidance.[7] It aims to support researchers, policy makers, practitioners, funders and journal editors in the funding, conducting and reporting of research on adaptation. The ADAPT study comprised of three work packages: (1) a systematic review of existing adaptation guidance[2] and scoping review of case examples of intervention adaptation[20]; (2) qualitative study using semistructured interviews to explore the understandings, perspectives and experiences of researchers, funders, journal editors, and policy and practice stakeholders; and (3) a Delphi expertise consensus exercise to scope the clarity of the definitions and constructs used in the guidance, explore and capture key debates, identify agreement on important adaptation processes, and ascertain areas where there is limited consensus.[21] These work packages formed part of the process to develop the guidance and the current study forms part of work package 2.

## METHODS

This paper reports on the semistructured interviews which were undertaken between April and September 2019 concurrently to inform the ADAPT study guidance. Participants were stakeholders with experience of intervention adaptation.

A case study research design was used in the first instance.[22–24] A case of adaptation was defined as a population health intervention that had previously been subjected to adaptation or was currently being adapted. For each of the cases, we aimed to interview a researcher involved in intervention adaptation and/or re-evaluation, patient and public involvement (PPI) contributors who were part of the intervention adaptation and where possible an associated decision-makers (eg, policy and/or practitioner stakeholder) who may have had experience of implementing the intervention in the new context. Although, in many cases, there was only one perspective represented per case. Funders and journal editors were not linked to specific cases but contributed to understanding of the wider evaluation context. As the study unfolded, it became increasingly challenging to recruit multiple and varied participants per case. Therefore, in many cases, there was only one perspective represented per case. In order to redress this shortcoming, more emphasis was placed on exploring diverse perspectives across different participants linked to different adapted interventions rather than comparing across cases.

### Recruitment and sampling

Researchers, policy makers and practitioners were initially identified through case examples of adapted interventions retrieved as part of the ADAPT systematic review[2] and scoping review.[20] The studies were included if they were a primary study describing an adaptation process and/or an evaluation of an evidence-informed intervention adapted to a new context, focused on public health and/or health service interventions, and were published from 2000 onwards. Studies were excluded if the intervention had been designed de novo for a specific context or examined clinical procedures, such as surgery. The 312 retrieved interventions were classified according to the socioecological domain where the theory of change primarily operated (mico, meso or macro); the contexts between which the intervention was transferred (eg, country to country or population to population within a country), study design (eg, effectiveness or feasibility) and outcomes (ie, favourable or unfavourable). The purpose of this was to achieve insight into variations in the nature of system disruptions (areas the intervention intends to target and enact change on), adaptations and adaptations processes, and how they might explain different outcomes. During the recruitment process, participants were emailed the information sheet and the consent form and asked to provide consent to take part in the study prior to the interview. All participants were given at least a week to consider their participation prior to their completion of the consent form.

**Table 1** Adaptation cases sample characteristics

| Stage of study | Participant (researcher/practitioner) | Type of intervention (macro/meso/micro) | Research design (feasibility study or RCT) | Target of intervention | Contextual transfer (country to country/population to population/setting to setting) | Evaluation outcome |
|---|---|---|---|---|---|---|
| Adaptation cases with two stakeholder perspectives | | | | | | |
| Completed | Researcher and practitioner | Meso | Feasibility | Diet and exercise | Policy to different settings | Infeasible |
| Adaptation cases with one stakeholder perspective | | | | | | |
| Completed | Researcher | Macro | Feasibility | Reproductive and child health | Country to country | Feasible |
| Completed | Researcher | Macro | Feasibility | Road traffic injury | Country to country | Feasible |
| Completed | Researcher | Meso | RCT | Addictions | Country to country | Effective |
| Completed | Researcher | Meso | Feasibility | Sexual health | Population to population | Feasible |
| Completed | Researcher | Meso | Feasibility | Sexual health | Population to population | Effective |
| Completed | Researcher | Meso | Feasibility | Hearing | Setting to setting | Feasible |
| Completed | Researcher | Micro | RCT | Parenting | Country to country | Effective |
| Completed | Researcher | Micro | RCT | Weight Loss | Population to population | Effective |
| Completed | Researcher | Micro | Feasibility | Diabetes prevention and management | Population to population | Feasible |
| Completed | Researcher | Micro | Feasibility | Smoking: cessation | Population to population | Feasible |
| Completed | Researcher | Micro | Feasibility | Mental health | Country to country | Feasible |
| Completed | Researcher | Micro | Feasibility | Childhood obesity | Setting to setting | Feasible |
| Completed | Researcher | Micro | Feasibility | Exercise | Population to population | Infeasible |
| Completed | Practitioner | Micro | Feasibility and RCT | Addictions | Setting to setting | Mixed |
| Completed | Practitioner | Micro | Feasibility and RCT | Addictions | Setting to setting | Mixed |
| In progress | Researcher | Meso | RCT | Lung health | Country to country | N/A |
| In progress | Researcher | Meso | RCT | Cancer | Population to population | N/A |
| In progress | Researcher | Meso | Feasibility | Weight loss | Country to country | N/A |
| In progress | Researcher | Micro | RCT | Diabetes prevention and management | Population to population | N/A |
| In progress | Researcher | Micro | RCT | Diabetes prevention and management | Population to population | N/A |
| In progress | Researcher | Micro | RCT | Diabetes prevention and management | Population to population | N/A |
| In progress | Researcher | Micro | Feasibility | Weight loss | Country to country | N/A |
| In progress | Researcher | Micro | Feasibility and RCT | Diet and exercise | Country to country | N/A |

All 23 primary researchers, who were recruited, were contacted, with the aim to snowball sample further stakeholders. This was largely ineffectual; this yielded three participants due to the age of some of the studies, therefore, additional recruitment strategies were used: expertise recommendation; advertising through the Involving People charity, which supports public and patient involvement in research; and Twitter promotion targeting the European Society for Prevention Research and the Society for Prevention Research. Funders were identified from international funding boards. Journal editors were identified from the relevant journals that published the case examples of adapted interventions.

A total of 37 participants were recruited to the study. The sample comprised of 23 researchers involved in the adaptation of 23 interventions (cases) (table 1).

The researcher participants conducted their work in the USA (n=12), UK (n=2), New Zealand (n=2), India (n=1), France (n=1), Germany (n=1), Spain (n=1), Italy (n=1), China (n=1) and Germany (n=1). Of the three practitioners, one practitioner was linked to one of the 23 interventions and two were recruited via expert recommendation. These practitioners had experience of adapting interventions for addictions b these interventions were not one of the 23 interventions included. Two of the practitioners conducted their work within the UK and one conducted their work in France. The study did not succeed in recruiting PPI representatives or policy makers. Six representatives from funding panels participated. They were based in the USA (n=1), UK (n=3), Germany (n=1) or had an international remit (n=1). The five journal editors represented global health (2) or

public health (3). Their primary publishing location was USA (2), Canada (1), countries across Europe (1) and Australia (1). Among the approached individuals that did not take part, invitees stated that the subject matter was not relevant to them (6), their workload was too high (2) or they did not respond after three follow-up contacts (64).

## Data collection

Interviews were conducted by two members of the research team (LC/HJL). Tailored topic guides were developed for each set of researchers and stakeholders, informed by the study research questions and emerging data from the systematic[2] and scoping reviews.[20] Guides were refined and confirmed with the wider study team prior to data collection. They were also reviewed as the interview progressed and no revisions were necessary. They considered the definition of intervention adaptation and related concepts; experiences of undertaking adaptation and re-evaluation, in addition to funding and reporting adaptation processes; and views on adaptation guidance development (see online supplemental appendix A). Interview length ranged from 40 to 75 min and were conducted via telephone or Skype. Interviews were audio-recorded and transcribed verbatim by a professional transcription company. Transcripts were reviewed for accuracy and anonymised.

## Data analysis

Four members of the research team (LC, DC, HJL, RE) analysed the data using the framework approach.[25] The three participant datasets (researchers and practitioners; funders; journal editors) were treated separately. Three different coding frameworks were then developed by the four researchers, using two interviews from each dataset which were randomly chosen. Each framework included both a priori codes and in vivo codes. The remaining data were coded by a single researcher. The frameworks evolved during analysis, with the new codes discussed and confirmed by the team, before being applied to previously coded data. To ensure reliability, 10% of the data was independently checked by a second researcher (RE/DC). Disagreements between researchers were resolved through discussion. NVivo V.10 supported data analysis and storage.

The four researchers charted coded data into the three separate framework matrices. Data within and across the matrices were compared and contrasted by two members of the research team (LC, RE) as part of the interpretative process of generating themes. To aid this process, visual maps were created. We created five overarching themes, each with a set of related subthemes: adaptation decision-making and processes, re-evaluation decision-making and processes; funding; publication; and recommendations for adaptation guidance. Overarching themes were presented to the wider ADAPT study team who suggested further refinements of subthemes. As the ADAPT Delphi consensus exercise progressed and areas of consensus

and disagreement emerged, we undertook additional analysis of the qualitative data to bring insight to these emerging perspectives.

## Patient and public involvement

This research was conducted without patient involvement. We involved policy and practice representatives with experience of intervention adaptation in qualitative interviews and our study advisory group.

## Reflexivity

LC and HJL conducted the interviews, and LC, HJL, DC and RE conducted the data analysis. At the time of analysis, LC and HJL were research associates with PhDs. DC was a research assistant with an MSc. RE was a senior lecturer with a PhD. All are experienced qualitative researchers who have received training in conducting interviews and thematic and framework analysis. None of the researchers apart from RE and HJL had a prior relationship before the study. RE and HJL had worked previously on studies together. The participants did not know the researchers prior to the study. The participants understood the researchers were conducting the interviews as part of the ADAPT study in order to explore their experiences of conducting adaptation studies. RE and HJL have a methodological expertise in adaptation which may have influenced their interview style and analysis of the data based on their extensive prior knowledge of the area. LC and DC were new to adaptation, but both have worked on process evaluations looking at context. Therefore, their focus on context may have influenced the interview style and analysis. The interviews were guided by topic guides developed by the wider team which will have negated some of the researcher bias. Ten per cent of the analysis was double coded to negate some of the bias of the researchers.

## RESULTS

The analysis generated four central themes. The first three relate to participants' experiences of and reflections on intervention adaptation (1) experience of involving stakeholders in the adaptation process; (2) negotiating the mismatch between the original context where the intervention was delivered and the new context; (3) deciding on the re-evaluation process. The final theme (4) reflects on participants' experiences of using adaptation frameworks, and their recommendations for future guidance development.

## Involving stakeholders

Participants foregrounded the importance of involving a diverse range of stakeholders (intervention developer, industry policy makers, implementers and organisations supporting delivery, and participants) throughout the adaptation process. The reasons for stakeholders' engagement were primarily related to them having more knowledge of how the intervention functions or the

characteristics of the new context, compared with those leading the process, who were often academics.

> Absolutely I mean Apple doesn't develop an iPhone without doing market research. We, as researchers and clinicians and doctors, you know, we have the knowledge from the textbook and from the theory and everything we've learnt, but we don't understand how to apply to a specific population without their knowledge and their expertise to teach us how it would be relevant for them. (P008 researcher micro in progress)

Stakeholder involvement was considered to be so imperative within the adaptation process, that funding representatives maintained that it was a central criterion applied by an assessment panel:

> They will always come together if there are key methodological flaws … that will come out very quickly and that includes things like the PPI involvement is simply not there, and that is something that they consider mission critical, and as I say. (P004 funder)

The practicalities of involving stakeholders was also noted as problematic. This could be due to potential conflicts of interest between stakeholders and researchers. For example, stakeholders could direct adaptations in ways that are under-researched. One practitioner discussed their experiences of working in a setting in which they felt the evidence base did not work within the context creating a conflict between experience and evidence-driven adaptations:

> So although it's sitting there saying this is the evidence base, you should be doing this, it just categorically doesn't work in our setting. So that's an example of how if you just applied the evidence base it just, it would be hopeless. (P022 practitioner)

This causes a conflict between ensuring that all contributions are supported by evidence, that is, low risk of bias versus changes being made based on stakeholder experience of the setting. This can cause issues when reviewers are reviewing the study and assessing how the adaptations have been justified. For example, one journal editor highlighted that many reviewers are not experts in adaptation and do not always assess the quality of PPI involvement:

> There's a lot more need for PPI stuff, there's a lot more need for doing more of the background work I'd say, the formative work to get input from key stakeholders and recipients … and that's the kind of stuff that a lot of reviewers don't even pick up or think about because they don't do the work. (P003 journal editor)

In addition, many stakeholders were noted as making contributions at different times across the adaptation process, often with differing opinions and different expressions of need. This made it difficult to undertake adaptation systematically, incorporating and balancing the 'stylistic differences' across the stakeholders about what should be done:

> But through just team discussion and supervision and peer supervision, we definitely have a kind of, if you like a general consensus that adaptations are necessary, and I think we all do that. But it's just the degree to which we do it, and exactly how we do it will vary from clinician to clinician. (P022 practitioner)

Stakeholder involvement within adaptation, as described by all the participants, is key as they can provide insight into how the intervention might function in the new context and what adaptations the new context may require. However, it was noted that there are multiple perspectives and differences of opinion and values about what to adapt and why, which can pose barriers to effective adaptation.

### Selecting the intervention and negotiating the mismatch between the original and the new context

Most data pertaining to participants' experiences of intervention adaptation centred on how to select the right intervention for the new context, how to decide if adaptation is necessary and if so which adaptations to undertake. Overall, there was a sense of tension between wanting to select the intervention based on evidence and ensuring it could be delivered with the resources and money available. However, in reality there are competing practical factors that need to be taken into account to guide decisions. For example, one researcher reflected on the issue of balancing the evidence base vs the practical aspect of ensuring that the intervention could be delivered in a low-income country with different resources.

> The most important thing I take first and foremost is the degree to which the evidence is available and is robust enough to adapt into different environments and whether it's adaptable. Whether the rigour and the tool's ability of the intervention may be suitable in high income and may not even be adaptable in low and middle income countries. (P023 researcher macro feasible)

Again, pragmatically, participants chose to use evidence-based interventions that were already embedded in the country as it had already achieved buy in among stakeholders and there were already the mechanisms in place to support delivery. Participants often selected interventions based on an awareness or prior relationships with the developers or evaluators as they had built a trusting and respectful working relationship:

> It is probably like most studies, I would love to say it was fully systematic!. … We chose it because they are very faith-based and we thought that would work but to be honest, a good bit of it was that these were two good investigators I knew. (P010 researcher micro in progress)

When it came to the process of modifying the intervention, participants reflected on how time-consuming and therefore complex adaptation could be. Some maintained that it could take up to a year to iteratively adapt the intervention, depending on the level of complexity involved. There was a clear sense that the current funding climate, which often subsumed adaptation into early phases of evaluation, did not permit the required time to fully undertake comprehensive adaptation:

It's very rare I guess, to get funding that is explicitly and exclusively for adapting a campaign. So that kind of funding mechanism is unusual, but it really gave us a chance to do things the right way. (P016 researcher meso infeasible)

In general, it was reported that there was a limited amount of time that could be funded to conduct the adaptation process. One funder commented that they only allow 6 months for this, which they felt was not sufficient time to conduct the adaptation process.

I mean probably a limitation of the system is that we have kind of a rule in that we'll only fund up to six months of adaptation work. (P003 funder)

Overall, the adaptation process is complex and involves balancing pragmatic decisions with decisions based on evidence, which researchers have been trained to prioritise. It requires time and a systematic approach to ensure a thorough process is undertaken; however, this is difficult if this process is not recognised by the funder and with no consistent and systematic approach to follow, at the time of data collection.

### Deciding on the re-evaluation process

Some re-evaluation was considered by most to be necessary following the introduction of an adapted intervention into a new context, although deciding on the nature and extent of new evaluation required was described as challenging. Participants discussed how they considered the utility of different study designs for re-evaluation and the complexity of deciding on an approach. A number of individuals suggested that feasibility testing, process evaluations and implementation studies are most relevant, given that the most pertinent research questions relate to mechanism of action.

Despite some indication of the rationale for different evaluation designs, in practice participants encountered numerous challenges to the conduct of a scientifically robust evaluation. While it was common for participants to state a preference for less resource intensive evaluation, on occasion they did acknowledge the importance in resolving uncertainty. It appears that this can lead to researchers and funders being at odds as researchers feel they can borrow strength from the existing evidence and skip steps. Whereas, the funder default may be still to expect evaluation as if it is a new and untested intervention.

It was forced upon us, I think it'd be true to say. [Laughs]. We'd decided, obviously, on an adaptation phase and, in fact, we wanted to go straight for a full trial. Because our views, you know, naivety galore, thought that this was a great programme from (name of place 3), and why not de-Anglify it, make some adaptations and pretty much roll straight out into full trial. Obviously, I think we probably had a feasibility, you know, internal pilot, I can't quite remember, actually, at our Stage 1 application. And they came back, saying, "No, no, we don't think you should go beyond feasibility phase." (P014 researcher micro in progress)

Centrally, participants reflected on the resource required for extensive re-evaluation, notably in terms of time and funding, which could not always be acquired:

We need theory building and we need that work, but I feel like with the limited funding that's available, particularly in the (name of place 1) and the drastic health conditions that we have, that we probably should start matching and integrating our efficacy trials with our effectiveness trials, that we develop things with an eye for sustainability and thinking about how to leverage the current resources that we have. (P008 researcher micro in progress)

There are clear challenges to re-evaluation which derive from a lack of certainty about how researchers make decisions about what type of evaluation to undertake and how funders make judgements about what to fund. There are merits to the different research designs, however, participants did not know which design was most suitable for their intervention and context and, at the time of data collection, there was no recommended systematic approach for how to make decisions about re-evaluating adapted interventions to use. Therefore, these findings identified a real need for guidance to inform the current uncertainty surrounding funding decisions and resources.

### Participants' experiences of using adaptation frameworks and recommendations for future guidance

Participants described limited awareness of adaptation frameworks, rarely mentioning their uptake. However, when mentioned, they were seen as important to conduct research in a systematic manner. In the absence of dedicated adaptation models, most participants drew on generic intervention development and evaluation guidance to support their decision-making processes. There were recognised limitations with existing adaptation frameworks and guidance. First, they were considered too long and time-consuming to be realistically applied given the resource constraints associated with the current funding climate. One participant followed the Map of the Adaptation Process,[11] which was deemed to be shorter compared with other frameworks, due to time constraints.

Yes, so I … we followed the Map of the Adaptation Process, right, that is more, it's a shorter version, and

it's still grounded in theoretical approaches…. Time was one of the factors, being cost effective was another, and we didn't have enough funding for a thorough and long adaptation process. (P012 researcher meso feasible)

Second, participants suggested that guidance can often be too conceptual, making it difficult to implement in real-world practice. In particular, there was a challenge in applying and tailoring generic, abstract thinking to the detailed specifics of the intervention they were working with:

It's so specific to each intervention, these things are so specific that it's really hard to pin them down, and to say well, to move from the concept to the actual practical side of things is quite difficult. I think that's probably the biggest challenge. (P005 researcher meso unfeasible)

Reflecting on these issues, participants expressed a number of recommendations for the development or future refinement of adaptation guidance. Some participants expressed a need for an overarching, systematic timeline of adaptation phases and re-evaluation approaches to allow for a common understanding across stakeholders of the adaptation and re-evaluation process:

I think it's always good to have a systematic kind of timeline in terms of when you should do stuff. (P019 researcher meso in progress)

In order to fully recognise the value of stakeholder involvement, participants stated that guidance also needed to target the full range of relevant stakeholders This can enhance buy in by ensuring that the guidance can be understood by different stakeholders and providing a process for how to involve them throughout the study:

…adaptation requires time and results and skills, and policy makers don't know that (laughs) at all. I think it's important to just have some guidelines or tools to let them understand, because I'd rather that it's one of your targets, but I think that's also the information you should give that's different for policymakers or practitioners or researchers. (P017 researcher meso effective)

Finally, there was suggestion for a checklist in terms of what to include when reporting adaptation processes in papers for publication. Participants talked about multiple influences in terms of publications such[26] as the time the researcher has, the type of paper that gets published and the need to accurately report the adaptation process.

So having like a very big and broad checklist of things to think about, and probably will be something that you have nothing to do with you, but at least you can follow that one, like a third guideline to see what other things that you need to report. (P001 researcher micro feasible)

This will aid publication of adaptation process papers as well as outcome papers.

## DISCUSSION

This qualitative study explored the real-world experiences of researchers, practitioners, funders and journal editors of conducting adaptation research. This work has highlighted a number of key challenges: (1) involving stakeholders throughout the adaptation process and how to integrate the evidence base with experience; (2) selecting the intervention and negotiating any incongruence between the original intervention and the new context; (3) the complexity and uncertainty of deciding on the re-evaluation process; and (4) participants' experiences of using adaptation frameworks in practice. These findings contributed to the ADAPT guidance[12] and address important gaps in our knowledge about the adaptation, implementation and re-evaluation of complex interventions in new contexts.

The participants repeatedly highlighted the importance of stakeholder involvement throughout the adaptation and re-evaluation process[14 17] as they provided an insight into the intervention's functioning or the features of the new context. However, there are challenges in coproduction research,[18] as raised by the participants, in terms of ensuring adaptation is conducted in a systematic and evidence-based manner. This uncertainty is echoed in the work of community-based participatory research in which it is challenging to anchor it in comprehensive theoretical framework.[27] Due to the importance of stakeholders, the participants stressed the need for the guidance to be accessible and presented in a way that helps to involve stakeholders.

Selecting the intervention and negotiating the mismatch between the original and the new context presented challenges for the participants. They reflected that the selection process was complicated as researchers wanted to base their decisions on the current evidence base; however, they also acknowledged that there were practical considerations that could compete with the evidence base or override it. For example, after reviewing all the evidence on an intervention, one might find that the evidence indicates a particular intervention is the most effective. However, it may be too resource intensive to be implemented within the new context.[28] Therefore, pragmatically it might be best to select an intervention that is already embedded in the country as it already has the mechanisms in place to support delivery and in addition has gained the buy in of stakeholders.[29] This balance between evidence and practice-based decisions is a consistent challenge throughout public health research and is an aspect that needs to be resolved to help bridge the gap between research and practice.[30–32] This is an unresolved area which, if left unaddressed, could impact the scientific merit of the selected intervention.[32] Therefore, guidance is needed to clarify the intervention selection process and bridge the research and practice gap.

Participants reported the re-evaluation process and the merits of different research designs. Overall, it was found that there is currently much uncertainty as to which design to choose. Researchers reported deciding that more extensive adaptation required a randomised controlled trial (RCT) (it was acknowledged that this design might not always be appropriate to assess the intervention, eg, a policy intervention at a macro level) to be conducted as there was greater uncertainty as to whether the intervention would remain effective. They also indicated that if the original intervention has had multiple RCTs already conducted showing effectiveness in the original context, they perceived that no pilot would be required during revaluation.[26 33] However, funders have recommended to researchers that pilot studies should be conducted as an initial re-evaluation stage.[34 35] Participants felt that there was a tension between these time requirements and the funding climate, at the time of data collection, which did not accommodate the required time to fully undertake comprehensive adaptation. Given the current funding climate and time to test feasibility and effectiveness, participants expressed a need for less resource-intensive evaluation.[34] To address this issue in part, it is important to place value on the information already existing for the intervention in its original context. This can aid decisions on whether a full evaluation is warranted prior to implementation.[36]

Overall, participants reflected that there were several challenges of using adaptation frameworks in practice. A number of adaptation frameworks have been developed in order to provide some guidance for this emerging field. However, while some aspects of good practice are clear, there are still areas on which there is no consensus on best practice.[2] Some frameworks were reported to be difficult to implement within real-world settings due to the oversimplified, list-like format which does not reflect the complex nature of the adaptation process.[37] Further to this, it was reported some frameworks were too time-consuming, leading to interpretation issues due to funding restrictions. Participants expressed a desire for guidance to take into account real-world challenges and for it to reflect the different time and funding availability.

There are practical challenges that have been raised by the participants within this study. This area is constantly progressing with emerging adaptation frameworks,[1 2 13 14] and now with the recently published ADAPT guidance, there is a need to assess how such guidance can help support these identified practical challenges going forward. As highlighted by the participants, there are limited resources and funding available, as well as a drive towards value for money. Therefore, adaptation can provide a cost-effective way of tackling the health needs of different settings,[11] with the right support and buy-in from funding organisations. Overall, there was a clearly expressed need for guidance from study participants. However, in this quickly evolving field, it is important to engage with how the guidance is being used and the nuance and diversity in perspectives on an ongoing basis.

## Study strengths and limitations

The primary limitation of this study was that the diversity of perspectives reflected in our data was limited by failure to recruit from some target groups. Although there were multiple attempts to recruit PPI and policy makers, we were unsuccessful. Therefore, in the process of developing case studies. We suspect that this issue with recruitment was due to the majority of the studies being completed, therefore, many of those involved in the study had moved onto other job so were uncontactable or did not have the time to take part. This is reflective of the nature of research culture in which people are contracted only for the duration of the project. We were unable to recruit policy makers after reaching out to a number of contacts due to the busy nature of their jobs. As such, their perspectives, which may contrast with the generated data, were not included. In addition, without the input from policy makers, the study lacks insight into how intervention adaptation is commissioned and resourced at a national and local level. Furthermore, while we aimed to sample people involved in a wide range of interventions, operating across the micro, meso and macro domains, we were only able to identify two macro-level interventions meeting our criteria for adaptation.[20] This may be a consequence of such interventions, notably national policies, not being explicitly framed as adaptations even when derived from principles and practices that are implemented elsewhere. Regardless of these limitations, the data did capture a diverse and nuanced range of perspectives in relation to intervention adaptation. It provided complementary data that contributed to and triangulated with the other ADAPT work packages and facilitated the production of comprehensive guidance for researchers on adaptation.[38]

## Practice implications

As a result of this study, there are a number of recommendations for conducting adaptation research. Participants identified that a systematic approach to adaptation and a checklist for publication was vital to ensure the intervention and its interaction with the context are adequately considered, while directing available resources to the most important areas of uncertainty, and that all proactive and responsive adaptations are captured and justified both before and after adaptation (researcher or practitioner led).[39–41] However, as this is a new and developing field, there is also a need for flexibility to allow for innovation within the field. It is also important for the adaptation process to be accessible and work for different stakeholders to ensure their involvement throughout.

## CONCLUSIONS

This study highlights the range of challenges experienced in funding, conducting and reporting research on intervention adaptation. This is partly due to uncertainty about the processes that should be undertaken, and the fact that, at the time of study conduct, frameworks to

support adaptation have only recently emerged. Moving forward, guidance on intervention adaptation, including the ADAPT guidance, may be helpful in systematising processes provided that they remain responsive to the local contexts. Therefore, there is a need to assess if the current ADAPT guidance, whose development was informed by the results of this study and published after data collection and analysis for this study took place, can provide clarity. There is also a need to assess and ensure that this guidance is not being too reductionist, as this is an emerging area which requires room to grow.[41] Future research to monitor how adaptation research evolves, particularly as the ADAPT guidance begins to be used in real-world practice, would improve knowledge and understanding. This learning will help to support further development and refinement of the guidance, ensuring that future iterations are responsive to the everchanging context of evaluation research.

**Author affiliations**
[1]Centre for Development, Evaluation, Complexity and Implementation in Public Health Improvement (DECIPHer), Cardiff University, Cardiff, UK
[2]Pettenkofer School of Public Health (PSPH), Institute for Medical Information Processing, Biometry and Epidemiology, LMU, Munchen, Bayern, Germany
[3]Nursing, Midwifery and Allied Health Professional Research Unit, University of Stirling, Stirling, UK
[4]Centre for Trials Research, Cardiff University, Cardiff, UK

**Acknowledgements** GM, PH, SM and JS: Conceptualisation, funding acquisition and writing—review and editing. RE: Conceptualisation, funding acquisition, formal analysis, methodology, project administration, supervision, visualisation, writing—original draft preparation, and writing—review and editing. LC: Data curation, formal analysis, project administration, visualisation, writing—original draft preparation, and writing—review and editing. HJL: Conceptualisation, funding acquisition, data curation, formal analysis and writing—review and editing. DC: Data curation, formal analysis and writing—review and editing. RE: is the guarantor.

**Funding** The ADAPT Study was funded by the MRC-NIHR Methodology Research Programme (MR/R013357/1). The project was undertaken with the support of The Centre for the Development and Evaluation of Complex Interventions for Public Health Improvement (DECIPHer), a UKCRC Public Health Research Centre of Excellence. Joint funding (MR/KO232331/1) from the British Heart Foundation, Cancer Research UK, Economic and Social Research Council, Medical Research Council, the Welsh Government, and the Wellcome Trust, under the auspices of the UK Clinical Research Collaboration, is gratefully acknowledged. The study was also supported by its successor, the Centre for Development, Evaluation, Complexity and Implementation in Public Health improvement, funded by Health and Care Research Wales from 2020. Peter Craig and Mhairi Campbell received funding from the UK Medical Research Council (MC_UU_12017-13) and the Scottish Government Chief Scientist Office (SPHSU13).

**Competing interests** None declared.

**Patient and public involvement** Patients and/or the public were not involved in the design, or conduct, or reporting or dissemination plans of this research.

**Patient consent for publication** Not applicable.

**Ethics approval** Ethical approval was provided by Cardiff University's School of Social Sciences Ethics Committee (Ref: SREC/3165). Participants gave informed consent to participate in the study before taking part.

**Provenance and peer review** Not commissioned; externally peer reviewed.

**Data availability statement** Data are available upon request.

**ORCID iDs**
Lauren Copeland http://orcid.org/0000-0003-0387-9607
Pat Hoddinott http://orcid.org/0000-0002-4372-9681
Jeremy Segrott http://orcid.org/0000-0001-6215-0870
Rhiannon E Evans http://orcid.org/0000-0002-0239-6331

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
