## [Reviewer comments · BMJ Open]

ARTICLE DETAILS

TITLE (PROVISIONAL)	Adapting population health interventions for new contexts: Qualitative interviews understanding the experiences, practices and challenges of researchers, funders and journal editors
AUTHORS	Copeland, Lauren; Littlecott, Hannah; Couturiaux, Danielle; Hoddinott, Pat; Segrott, Jeremy; Murphy, Simon; Moore, Graham; Evans, Rhiannon

VERSION 1 – REVIEW

REVIEWER	Ali Mehryar Karim Bill & Melinda Gates Foundation, Global Development
REVIEW RETURNED	09-Sep-2021

GENERAL COMMENTS	The paper sought to understand the experiences of the major categories of stakeholders engaged with implementing and documenting adaptive public health interventions. Regretfully I stopped reviewing the article after reading the methods section mainly for two reasons: 1) if the audience is not adept in adaptive research, they will need to read many of the referenced articles to get a basic understanding of the theoretical framework and the study methodology. 2) The article summary did not provide me the curiosity to read the entire paper.
--

REVIEWER	Severin Rakic Public Health Institute of the Republic of Srpska, Centre for Health System Development and International Cooperation
REVIEW RETURNED	12-Oct-2021

GENERAL COMMENTS	Dear authors, My suggestions for revision of the manuscript are the following: OVERALL: 1. The manuscript might benefit from additional proofreading prior to resubmission. There are unclear sentences (e.g. line 92-93), missing words (e.g. line 94) and typos (e.g. line 241). Results and discussion sections might particularly benefit from editing that would strengthen main messages in the sections. TITLE: 2. The sample included no policy makers. It is not correct to have them in the title. ARTICLE SUMMARY: 3. You reflected on methodology and limitations, which is less relevant than presenting key findings and conclusions. Summary needs to be revised. 4. What does the PPI stands for (not defined throughout the manuscript)
--

	BACKGROUND: 5. A clear definition of population health interventions that you used throughout the ADAPT study is missing. METHODS: 6. It is not clear which type of multiple case study was used. It seems that you considered each included study to be a separate case study, but the results are then aggregated from different perspectives - which makes things unclear, particularly how was the analysis across cases handled. 7. The manuscript does not provide details on countries in which researchers and practitioners conducted their work. It is difficult to consider relevance of the manuscript for different settings without this information. 8. A clear explanation is missing on why you have chosen to include different groups in the sample. 9. Lines 143-146: Not clear whether the 3 practitioners were eventually included or not. 10. Were the interview guides piloted or not? 11. Inclusion and exclusion criteria for the studies are not provided. Based on the table 1, it looks like you focused on NCDs, mental health, road traffic accidents, and addictions, excluding communicable diseases. 12. The manuscript itself does not provide information on how was the interview consent handled. The information is part of additional files in appendix. TABLE 1: 13. The title of table is inadequate. It refers to participants in the sample, while the table provide information on included studies. 14. No need to list policy makers in the column headings. 15. What does "feasibility" means in terms of research design? Is it feasibility study? Provide explanation of RCT abbreviation. 16. "Intervention outcome measurement" is not adequate column heading. The column does not provide outcomes or measurement types, but the main topics of the study. RESULTS: 17. Some of the citations might be shortened, to focus them on the key points that illustrate what was explained in the text. Uncleared parts (e.g. line 338) should not be provide in citations. Discussion can be expanded for the words gained by shortening the citations. 18. Lines 270-290 discuss selection of the interventions. It is not clear why this has been included under title 3.2. Shouldn't selection of the interventions come before the required modifications are negotiated? 19. Line 300: Identification number might need to be corrected. 20. Line 358: subheading should be numbered 3.4 21. It would be useful to have more reflections on differences between different studies (RCT vs. feasibility studies) and between views of different perspectives (funders vs. researchers vs. practitioners vs. editors) included in the results. DISCUSSION: 22. The discussion needs to be expanded and better structured. To make it consistent with the rest of the manuscript, it might be structured around main themes (results have 4 main themes, but discussion refers to 3) OR main perspectives (taken into account title of the manuscript, this is what a reader would expect). 23. Practice implications are missing. It is not sufficient to simply refer a reader to the new ADAPT guidelines. Could differences between RCT and feasibility studies be discussed here? 24. Discussion should be better linked with existing literature, however limited the literature might be.
--	---

REVIEWER	Zahi Jurdi Medical University of South Carolina, Healthcare Leadership & Management
REVIEW RETURNED	15-Jan-2022

GENERAL COMMENTS	Thank you for the opportunity to review your manuscript. I would suggest proof reading the paper again as there are several gramatical errors. Your findings are quite value adding.
---

REVIEWER	George Karani Cardiff Metropolitan University School of Health Sciences, Occupational & Environmental Public Health
REVIEW RETURNED	25-Jan-2022

GENERAL COMMENTS	The subject of the paper is relevant, but I have some concerns that I would like to be addressed.  1. The lack of data from funders and PPI is indicated and highlighted as study limitation, but I would have liked a more detailed explanation why they were not able to recruit these two groups. 2. line 53-..'and countries.' these refer to three countries out of how many?. 3. Participants indicated line 54,55. Any bias as one group had considerable high n?Data analyses used TWO interviews from each data set from three stake holder sets, line 163. Selection criteria of these two interviews? 4.line 125 and 135- -became increasing challenging to recruit participants and also curious why method to recruit subjects was 'ineffectual' and would want further clarification to aid future researchers on subject. 5. Non response from individuals approached , line 150 and 151 is a concern. Of those (64) who did not respond, was there an attempt to re-contact them? 6. Re-check line 153 and 181 as to who conducted interviews and the data analysis. 7. Not sure the relevance on gender of researchers , line 181, 182. 8. Please re-check document and make sure acronyms given in full the first time used 9.Why does table 1, line 194 include only details of researchers and 2 practitioners? 10.re-check line 300 PO16 details as given. 11.I would have wanted a further discussion on differences and similarities from the views of the stakeholders in the study and include a table. I would wanted the author based on literature review completed in part 1 of the ADAPT project line 109, to discuss views / impact of funders and PPI on the study presented. 12. Would it be feasible to remove from paper title and abstract stakeholders who were not successfully recruited to the study?
---

VERSION 1 – AUTHOR RESPONSE

Reviewer's comment	Response
Reviewer 1	
Regretfully I stopped reviewing the article after reading the methods section mainly for two reasons: 1) if the audience is not adept in	We have addressed the issue of understanding by adding further explanation of the research design within the method section. We have

adaptive research, they will need to read many of the referenced articles to get a basic understanding of the theoretical framework and the study methodology. 2) The article summary did not provide me the curiosity to read the entire paper.	provided further information as to what is classified as a case and how we selected the case studies. See lines 133 to 137. We have also added definitions within the introduction to aid in the understanding of adaptation. See lines 88 to 93.
Reviewer 2	
The manuscript might benefit from additional proofreading prior to resubmission. There are unclear sentences (e.g. line 92-93), missing words (e.g. line 94) and typos (e.g. line 241). Results and discussion sections might particularly benefit from editing that would strengthen main messages in the sections.	The manuscript has been further proofread and the required changes made.
Title: The sample included no policy makers. It is not correct to have them in the title.	Policy makers have been removed from the title.
Summary You reflected on methodology and limitations, which is less relevant than presenting key findings and conclusions. Summary needs to be revised.	This was mislabelled as summary and has now been given the correct title of 'strengths and limitations of this study' therefore the limitations covered are appropriate. See line 74.
What does the PPI stands for (not defined throughout the manuscript)	This has now been defined in the summary and manuscript. See line 134.
Background: A clear definition of population health interventions that you used throughout the ADAPT study is missing.	The definition of population health has been added. See lines 91 to 93.
Methods It is not clear which type of multiple case study was used. It seems that you considered each included study to be a separate case study, but the results are then aggregated from different perspectives - which makes things unclear, particularly how was the analysis across cases handled.	As stated in line 131 a case was defined as a population health intervention, and we sought to recruit different stakeholders to each case. However as stated on line 136 recruitment to each case was challenging leading to one stakeholder per case. Therefore, the analysis was changed to looked at perspectives across different stakeholders rather than cases. We have added some clarity to this section to address this comment. See lines 131 to 142
The manuscript does not provide details on countries in which researchers and practitioners conducted their work. It is difficult to consider relevance of the manuscript for different settings without this information.	The details on the countries of the researchers and practitioners have now been added. See lines 169 to 171
A clear explanation is missing on why you have	We have now added an explanation as to why we

chosen to include different groups in the sample.	sampled the different groups of cases. See line 153 to 155.
Lines 143-146: Not clear whether the 3 practitioners were eventually included or not.	We have now clarified this, that they were participants. See line 161.
Were the interview guides piloted or not?	The interview guides were not formally piloted however they were reviewed by the wider research team and were reviewed as interview progressed to assess the need for revisions. See line 185.
Inclusion and exclusion criteria for the studies are not provided. Based on the table 1, it looks like you focused on NCDs, mental health, road traffic accidents, and addictions, excluding communicable diseases.	The inclusion and exclusion criteria have now been added to the methods which demonstrates the criteria for study selection. See lines 145 to 150.
The manuscript itself does not provide information on how was the interview consent handled. The information is part of additional files in appendix.	The consent process has now been added to the recruitment and sampling section. See lines 155 to 158.
TABLE 1: The title of table is inadequate. It refers to participants in the sample, while the table provide information on included studies.	I have changed the title to reflect that it is the sample characteristics for the cases rather than the participants. See table heading
No need to list policy makers in the column headings.	Policy maker has been removed. See table heading
What does "feasibility" means in terms od research design? Is it feasibility study? Provide explanation of RCT abbreviation.	This has been clarified as feasibility study within the heading of the table. RCT has also been explained as randomised control trial. See table heading
"Intervention outcome measurement" is not adequate column heading. The column does not provide outcomes or measurement types, but the main topics of the study.	This heading has been changed to target of the intervention to reflect that the topics are the public health areas the intervention aims to target. See table heading
Results: Some of the citations might be shortened, to focus them on the key points that illustrate what was explained in the text. Uncleared parts (e.g. line 338) should not be provide in citations. Discussion can be expanded for the words gained by shortening the citations.	The citations have been reduced in length to ensure that the keys points illustrated within them are brought to the forefront. Clarity has been provided to line 340 by rearranging the sentence. This has allowed the discussion to be further expanded.
Lines 270-290 discuss selection of the interventions. It is not clear why this has been included under title 3.2. Shouldn't selection of the	The title of 3.2 has now been changed to reflect that this section also includes selecting the intervention as well as looking at the differences

interventions come before the required modifications are negotiated?	in contexts. It therefore covers the section of the intervention first and moves onto the modifications. See line 291.
Line 300: Identification number might need to be corrected.	The identification number for this participant has now been changed to ensure consistency with the other identification numbers See line 322
Line 358: subheading should be numbered 3.4	This subheading number has been changed to 3.4.
It would be useful to have more reflections on differences between different studies (RCT vs. feasibility studies) and between views of different perspectives (funders vs. researchers vs. practitioners vs. editors) included in the results.	There was disagreement across the different stakeholders when it came to involving stakeholders as researchers prioritised adaptation based on evidence versus stakeholder making adaptation based on experience (see involving stakeholders section). In addition to this there was a tension identified between researchers who believe that the adaptation process takes time and is complex versus the funders who currently limit the time and funding allocated to the adaptation process (see intervention selection and mismatch section). There was also disagreement between researchers and funders in terms of re-evaluation in which researcher felt it was possible to borrow strength from existing research while funders wanted to intervention to tested again as if it was a new intervention (see re-evaluation section). On further investigation between the studies defined as feasibility vs RCT there was no difference in perspective between these groups. Many of the researchers had similar experience of the adaptation and its complex nature.
DISCUSSION: The discussion needs to be expanded and better structured. To make it consistent with the rest of the manuscript, it might be structured around main themes (results have 4 main themes, but discussion refers to 3) OR main perspectives (taken into account title of the manuscript, this is what a reader would expect).	The structure of the discussion has been amended to reflect the 4 main themes that were identified in the results. Each theme is clearly signposted and the challenges within each theme discussed.
Practice implications are missing. It is not sufficient to simply refer a reader to the new ADAPT guidelines. Could differences between RCT and feasibility studies be discussed here?	A new section on practice implications has now been added to the discussion section. See 3.2 practice implications.

Discussion should be better linked with existing literature, however limited the literature might be.	Addition literature has been added to the discussion.
Reviewer 3	
Thank you for the opportunity to review your manuscript. I would suggest proof reading the paper again as there are several grammatical errors.	Thank you so much for reviewing the manuscript. It has now been thoroughly proofread and all errors and typos corrected.
Reviewer 4	
The lack of data from funders and PPI is indicated and highlighted as study limitation, but I would have liked a more detailed explanation why they were not able to recruit these two groups.	We were unable to recruit policy makers and PPI representation. This challenge has been further explored within the strengths and limitations section in which we talk about the busy nature of policy makers and that many of the stakeholders associated with the studies had moved into other projects or were uncontactable. See line 489- 491
line 53-..'and countries.' these refer to three countries out of how many?.	We looked to samples across as many countries as possible to capture low, middle and high income countries. The countries in which the studies took place have now been outlined within the recruitment and sampling section. This gives the number of studies and the number of counties sampled. See line 169-171
Participants indicated line 54,55. Any bias as one group had considerable high n?Data analyses used TWO interviews from each data set from three stake holder sets, line 163. Selection criteria of these two interviews?	Although there are a larger number of researchers recruited this represents our sampling strategy in which we aimed recruit cases across the socio-ecological domain where the theory of change primarily operated (mico, meso or macro); the contexts between which the intervention was transferred (e.g. country to country or population to population within a country); study design (e.g. effectiveness or feasibility); and outcomes (i.e., favourable or unfavourable). Therefore, the number represents the variation across the adaptation case sample. The interviews selected to build the framework were selected at random which has now been added to the manuscript. See line 195
line 125 and 135- -became increasing challenging to recruit participants and also curious why method to recruit subjects was 'ineffectual' and would want further clarification to aid future researchers on subject.	We have added an explanation as to why recruitment was challenging. We believe due to the age of the studies many of those involved in the study had moved on to other jobs and were no longer contactable or did not have the time to

	take part. See line 489- 491
Non response from individuals approached , line 150 and 151 is a concern. Of those (64) who did not respond, was there an attempt to re-contact them?	We attempted to recontact those non responders 3 times allowing them time to reply. This has been added to the recruitment and sampling section. See line 180
Re-check line 153 and 181 as to who conducted interviews and the data analysis.	This has been checked and the break down of who did which aspect has been updated in the reflexivity section. See line 211
Not sure the relevance on gender of researchers , line 181, 182.	Gender has now been removed from this section.
Please re-check document and make sure acronyms given in full the first time used	Acronyms have been checked and written in full the first time they were used.
Why does table 1, line 194 include only details of researchers and 2 practitioners?	The table indicates the one adaptation case in which we recruit multiple stakeholders which is why it details that there is a researcher and practitioner associated with that case. The following cases have one stakeholder associated with them as indicated in the table.
I would have wanted a further discussion on differences and similarities from the views of the stakeholders in the study and include a table. I would wanted the author based on literature review completed in part 1 of the ADAPT project line 109, to discuss views / impact of funders and PPI on the study presented.	There was disagreement across the different stakeholders when it came to involving stakeholders as researchers' prioritised adaptation based on evidence versus stakeholder making adaptation based on experience (see involving stakeholders section). In addition to this there was a tension identified between researchers who believe that the adaptation process takes time and is complex versus the funders who currently limit the time and funding allocated to the adaptation process (see intervention selection and mismatch section). There was also disagreement between researchers and funders in terms of re-evaluation in which researcher felt it was possible to borrow strength from existing research while funders wanted to intervention to tested again as if it was a new intervention (see re-evaluation section). In terms discussing the impact of the literature review in the funders and PPI I am sorry I do not understand the comment. We did not have any PPI participants in the study. In terms of the funders the literature review was a separate section of the study that was not published prior

	to the interviews, and we did not share the results with the funders therefore it would not have impacted their views.
Would it be feasible to remove from paper title and abstract stakeholders who were not successfully recruited to the study?	Policymakers has been removed from the title and abstract.

VERSION 2 – REVIEW

REVIEWER	Severin Rakic Public Health Institute of the Republic of Srpska, Centre for Health System Development and International Cooperation
REVIEW RETURNED	20-Jul-2022

GENERAL COMMENTS	Thank you for addressing the comments and accepting the suggestions.
--

REVIEWER	Zahi Jurdi Medical University of South Carolina, Healthcare Leadership & Management
REVIEW RETURNED	31-Jul-2022

GENERAL COMMENTS	Thank you again for the opportunity to review this paper. Excellent job with the proofreading/edits. The paper is now free of grammatical errors. Perhaps, a follow-up study could include the perspective of policy makers as their input, experiences, and expertise could be quite value-adding to this area of research.
--

REVIEWER	George Karani Cardiff Metropolitan University School of Health Sciences, Occupational & Environmental Public Health
REVIEW RETURNED	09-Aug-2022

GENERAL COMMENTS	The manuscript reads well and all concerns / suggestions raised by me have been fully addressed. Well done Team.
---